# Spatial heterogeneity enhance robustness of large multi-species ecosystems

**Susanne Pettersson** * , **Martin Nilsson Jacobi**

Department of Space, Earth and Environment, Chalmers University of Technology, Gothenburg, Sweden

* susannep@chalmers.se

**Data Availability Statement:** Software is available from the Zenodo database (DOI 10.5281/zenodo. 5155722).

**Funding:** The author(s) received no specific funding for this work.

## Abstract

Understanding ecosystem stability and functioning is a long-standing goal in theoretical ecology, with one of the main tools being dynamical modelling of species abundances. With the help of spatially unresolved (well-mixed) population models and equilibrium dynamics, limits to stability and regions of various ecosystem robustness have been extensively mapped in terms of diversity (number of species), types of interactions, interaction strengths, varying interaction networks (for example plant-pollinator, food-web) and varying structures of these networks. Although many insights have been gained, the impact of spatial extension is not included in this body of knowledge. Recent studies of spatially explicit modelling on the other hand have shown that stability limits can be crossed and diversity increased for systems with spatial heterogeneity in species interactions and/or chaotic dynamics. Here we show that such crossing and diversity increase can appear under less strict conditions. We find that the mere possibility of varying species abundances at different spatial locations make possible the preservation or increase in diversity across previous boundaries thought to mark catastrophic transitions. In addition, we introduce and make explicit a multitude of different dynamics a spatially extended complex system can use to stabilise. This expanded stabilising repertoire of dynamics is largest at intermediate levels of dispersal. Thus we find that spatially extended systems with intermediate dispersal are more robust, in general have higher diversity and can stabilise beyond previous stability boundaries, in contrast to well-mixed systems.

## Author summary

One of the major challenges facing humanity is the fragmentation of wildlife habitats and decline in biodiversity due to human land-use practices and need for resources. We need to find ways to combine human prosperity with biodiversity conservation. To achieve this a solid understanding of ecosystem stability and functioning is paramount. One way to gain such insight is to find limits when we expect species to go extinct or ecosystems to collapse by simulations of interacting species populations. Many such stability limits have been found theoretically the last decades, but for simplification of modelling, studies often exclude that ecosystems are spread out in space. Here, we explicitly include space and thus allow for dispersal and spatial heterogeneity (local differences) in species abundances. We

**Competing interests:** The authors have declared that no competing interests exist.

find that for an ecosystem with the possibility of local spatial heterogeneity, the repertoire of the system's dynamical behaviour increases dramatically. This increase in possibilities increases system robustness, enables limits previously marking extinction or collapse to be crossed without any remarkable change in global species abundances, and increases biodiversity. Thus we elucidate an additional mechanism pointing to spatial heterogeneity as crucial for ecosystem stability. We find intermediate dispersal as the most favourable for robustness and diversity of ecosystems since they display the largest repertoire of dynamical behaviour.

## Introduction

There are many ways to think about and represent ecosystem functioning and stability, one prominent direction in mathematical ecology is the search for limits of stability in dynamical species population models in terms of some specified parameters or characteristics of the system. To this end Generalised-Lotka-Volterra (GLV) dynamics and modifications, with for example higher order interaction [1] (e.g. third species modifying how one species interacts with another) as well as many examples of more advanced interaction functions between species have been used. With a proliferation of stability results both confusing and enlightening, such as the role of diversity for stabilising ecosystems [2–4] or to what extent properties such as modularity [5] or nestedness [6] in interactions act to stabilise or destabilise or are aspects of the same property [7].

Specifically the question whether diversity (used as synonymous with species richness) and/ or strength of interaction between species act to destabilise or not, has drawn a lot of attention since the first mathematical formulations of large ecosystem stability with May's paper in the 1970s [2]. The conclusions are not unanimous but a majority of studies find that at a certain level of diversity (or interaction strengths) the system will no longer have a stable equilibrium solution [8]. This means that ecosystems cannot sustain too high diversity, variation or mean of interaction strengths and can possibly collapse when interactions are increased if for example an ecosystem's available space is reduced. Studies have also located limits to feasibility (abundance >0 for all species) where higher interaction strengths lead to extinctions (loss of feasibility) [9, 10], and systems vulnerable to perturbations in structural features (such as growth rates, or interaction strengths) [11, 12]. With even higher interaction strengths, abundances can start oscillating or the system collapses to a significantly smaller subset of species [9]. This latter limit is the classical limit recognised as a drastic change in system characteristics and many times referred to as collapse.

A prominent model indicating diversity as destabilising is the original Generalised-Lotka-Volterra model. The GLV equations have again become widely used for studies of large ecological communities to investigate generic properties of complex ecosystems using relatively few parameters [13, 14]. Although many insights have been gained by this approach, one of the many simplifications of the GLV model is the lack of space. This simplification leans on the assumption that a spatial average of both the interactions among species and the species abundances is sufficient to capture the dynamics and stability aspects of an ecosystem.

There are on the other hand studies specifically taking space into account, studying for example pattern formation where the diversity is small [15–17], space influence on interaction structure [18] and the spread of specific species [19]. Another type of spatial models are meta-communities. A meta-community is a collection of ecological communities in space connected by dispersal. Most meta-community studies find that intermediate dispersal is favourable for high diversity and robustness [20, 21], although systems under investigation comprised of very

few species. It is also known that high dispersal can synchronise fluctuations in population abundances in different habitats, increasing the likelihood of extinctions [20–23]. The mechanism behind both these results is the presence or absence respectively of a buffering effect between the local communities. Other meta-community studies have found meta-communities exhibiting macro-ecological patterns at the edge of collapse [24] and investigated local stability in the feasible domain [25]. A feature that still holds in the latter spatial stability analysis is that there is a boundary of radical shift and loss of stability.

Recent studies using the GLV equations with dispersal have found that spatial extension enables crossing of stability boundaries and higher diversity in comparison to spatially unresolved systems. These features rely on chaotic dynamics [26, 27] and spatial heterogeneity in interactions [26]. Using a spatial gradient representative of abiotic factors, [28] has shown that the level of dispersal influences the abruptness in species composition between neighbouring communities.

Motivated by the large literature on stability limits, structural stability (robustness) and recent interest in spatial extension, in this paper we study a GLV model with diffusion on a lattice with no differentiation between spatial locations such as spatial heterogeneity in interactions, carrying capacities, or intrinsic growth rates. In contrast to spatially unresolved and spatially differentiated models, this formulation lifts the assumption of average species abundances and allows us to investigate the bare consequences of spatial extension. We re-investigate stability limits, robustness and diversity for varying amounts of dispersal.

## Theory and methods

The version of the classical GLV model we will use as a base is stated below

$$\frac{d\phi_i}{dt} = \phi_i r_i \left( 1 - \frac{\phi_i}{K_i} \right) + \sigma \phi_i \sum_{j=1}^{N} A_{ij} \phi_j, \tag{1}$$

where $\phi_i$, $r_i$ and $K_i$ are species abundances, intrinsic growth rates and carrying capacities for species $i$ respectively. The web of interactions between species is represented by $A_{ij}$, a $N \times N$ matrix with a density of interactions, $c = 0.5$, and whose non-zero entries are drawn from a normal distribution with negative mean, $\mu = -0.5$ and variance one. This means all types of interactions (trophic, mutualistic, competitive etc.) are included although there is a larger percentage of competitive $(-, -)$ and amensialistic $(-, 0)$ interactions. The parameter $\sigma$ becomes the standard deviation (s.d.) of the interaction strengths and is often used as a tuning parameter and proxy for complexity. An increase in interaction strengths means an increase in complexity.

For small $\sigma$ in systems governed by Eq 1 with diversity $N$ the system will settle in a unique stable fixed point with all $N$ species present. Under increases in $\sigma$ (increasing complexity) at a certain threshold feasibility is lost. With continued increase beyond this threshold to stay in a stable fixed point species will successively go extinct. Further increase in $\sigma$ eventually pushes the system across the final stability boundary and the system transitions to either oscillations, chaos or a fixed point with a substantial loss of species. This latter limit has many times been referred to as collapse. The region between the two boundaries is structurally unstable meaning a small perturbation in parameters ($r_i$, $\sigma$, $c$ etc.) lead to qualitative change, in effect species extinctions [9]. This region can also have multiple stable fixed points with differing patterns of extant (non-extinct) species [29].

We introduce a spatial dimension into the GLV model by setting up a grid (or line) where all grid-points have the same interaction matrix. This in effect, sets the same maximum amount of species with the same interspecies interactions at every grid-point but allows for the possibility of different dynamics and local population abundances. The grid-points are

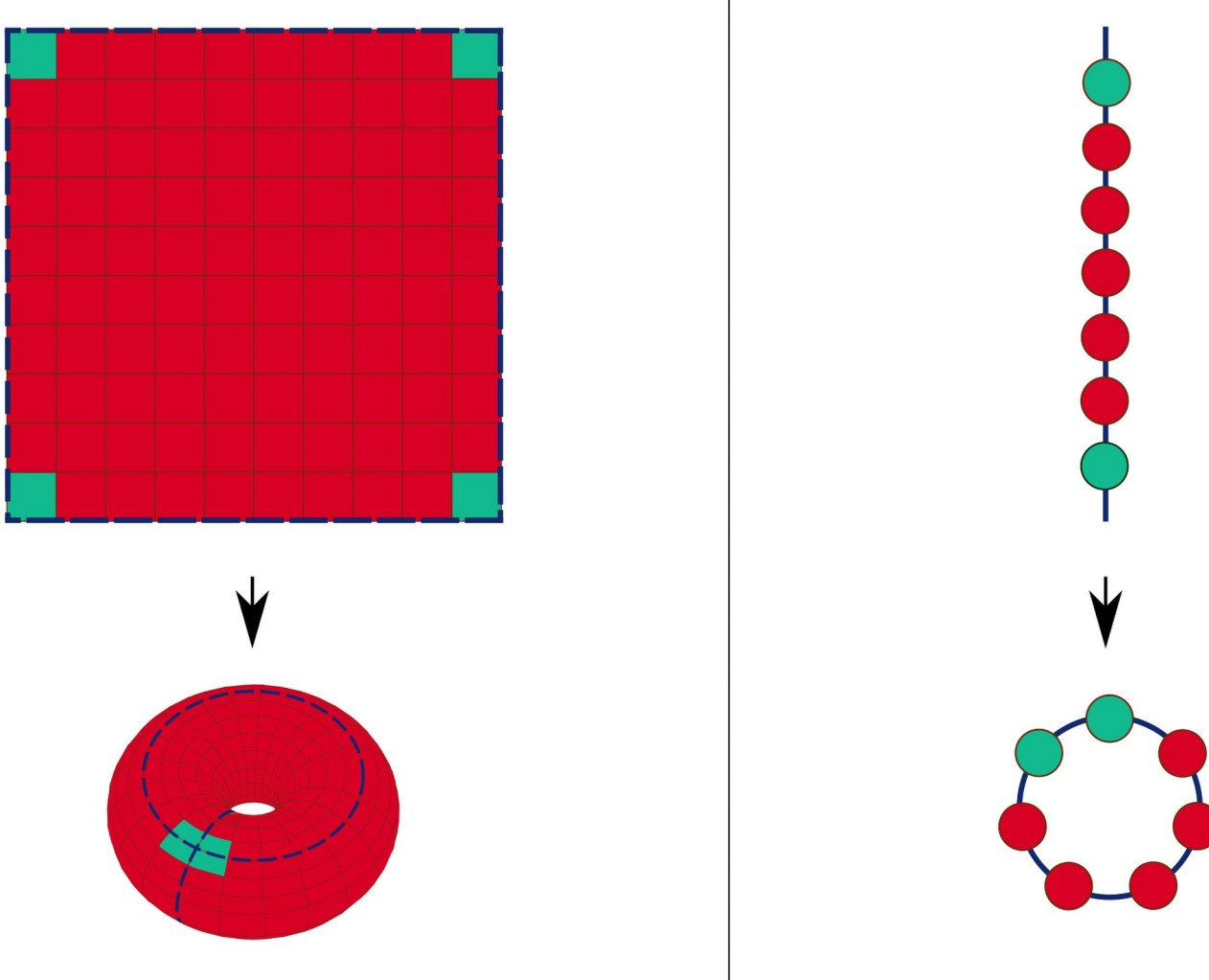

**Fig 1. Schematic grid and line.** The figure shows schematic pictures of how the grid is formed like a torus (to the right) and how the line is formed like a ring (to the left). This is done to simulate an ecosystem situated in a large connected space and to minimise boundary effects.

connected by diffusion, representing the dispersal of species in space. This can be seen as for example animals colonising areas where there is a larger abundance of prey per predator or plants dispersing to areas where there are less competitors (more available resources).

To avoid effects from the boundary of the grid we use periodic boundary conditions (shape the grid as a torus in two-dimensional space or ring in one-dimensional space) as shown in Fig 1.

The GLV equations in continuous space with diffusion are

$$
\begin{aligned}
\frac{\partial \phi_i(\mathbf{x}, t)}{\partial t} \quad &= r_i \phi_i(\mathbf{x}, t) \left( 1 - \frac{\phi_i(\mathbf{x}, t)}{K_i} \right) \\
&+ \sigma \phi_i(\mathbf{x}, t) \sum_{j=1}^{N} A_{ij} \phi_j(\mathbf{x}, t) \\
&+ D_i \sum_{p=1}^{2} \sum_{q=1}^{2} \frac{\partial^2 \phi_i(\mathbf{x}, t)}{\partial x_p \partial x_q},
\end{aligned}
\tag{2}
$$

where $\phi(\mathbf{x}, t)_i$ is the relative abundance of species $i$ which now depends on both spatial location $\mathbf{x}$ and time $t$. The diffusion rates for species $i$ are $D_i$, which are different for all species but same for all grid-points for the same species. The $D_i$ are random variables drawn from uniform distributions with means $\mu_D = 10^a$ with a, a parameter we varied from -5 to 1 in steps of 1. The different $D_i$ for different species is to represent that species have varying dispersal rates, although all the $D_i$ are kept at the same order of magnitude for each run by a standard deviation proportional to the mean $\sigma_D = b\mu_D$, with $0.3 < b < 0.4$, such that the diffusion intervals for different means are non-overlapping. Since we wish to use a grid we discretise the spatial dimension of Eq 2 using the a discrete Laplace operator in two or one dimension given as

$$
\begin{aligned}
\frac{\partial^2 \phi_i(\mathbf{x}, t)}{\partial x_p \partial x_q} &\approx \frac{\phi_{i,\alpha+1\beta} + \phi_{i,\alpha-1\beta} + \phi_{i,\alpha\beta+1} + \phi_{i,\alpha\beta-1} - 4\phi_{i,\alpha\beta}}{h^2} \\
\frac{\partial^2 \phi_i(x, t)}{\partial x^2} &\approx \frac{\phi_{i,\alpha+1} + \phi_{i,\alpha-1} - 2\phi_{i,\alpha}}{h}
\end{aligned}
\tag{3}
$$

where $\alpha$ and $\beta$ are grid indices and the denominator $h$ is the "distance" between patches set to 1. The resulting dynamical equation for species $i$ in grid-point $(\alpha, \beta)$ in a two dimensional grid is thus

$$
\begin{aligned}
\frac{\partial \phi_{i,\alpha\beta}}{\partial t} &= r_i \phi_{i,\alpha\beta} \left( 1 - \frac{\phi_{i,\alpha\beta}}{K_i} \right) + \sigma \phi_{i,\alpha\beta} \sum_{j=1}^{N} A_{ij} \phi_{j,\alpha\beta} \\
&\quad + D_i(\phi_{i,\alpha+1\beta} + \phi_{i,\alpha-1\beta} + \phi_{i,\alpha\beta+1} \\
&\quad + \phi_{i,\alpha\beta-1} - 4\phi_{i,\alpha\beta}).
\end{aligned}
\tag{4}
$$

The dispersal in our model is thus only between nearest neighbour grid-points.

Real ecosystems are often situated in either two- or three-dimensional space. In most of our simulations on the other hand we used one-dimensional space to reduce the computational load and effect of boundaries. This was acceptable since we noticed no qualitative difference in results between two- and one-dimensional simulations.

To analyse the dynamics of the model, we screen each grid-point for fixed points and oscillations in species abundances. If there are oscillations we measure amplitudes and the synchronisation of local species abundance oscillations between grid-points. The synchronisation of dynamics we measure with the maximum phase-shift in oscillations between the same species in different grid-points, in effect the maximum difference between arguments of the Fourier transform at the dominant oscillation frequency. When the mean maximum phase shift of all extant species is zero the grid is synchronised, when $\pi$ radians (180°) completely unsynchronised.

## Results

With the addition of a connected space we add the possibility of different dynamics or solutions, and timing of dynamics in different grid-points. Spatial heterogeneity (local species abundance differences) because of different dynamics is only possible when $\sigma$ is large enough (larger than the feasibility boundary) so that different solutions of the equations are available in the grid-points. In this case the system can end up with multiple fixed points, oscillatory patterns, or a combination of fixed points and oscillations. The same oscillatory dynamics can also give rise to spatial heterogeneity by phase-shifts in local species abundances between

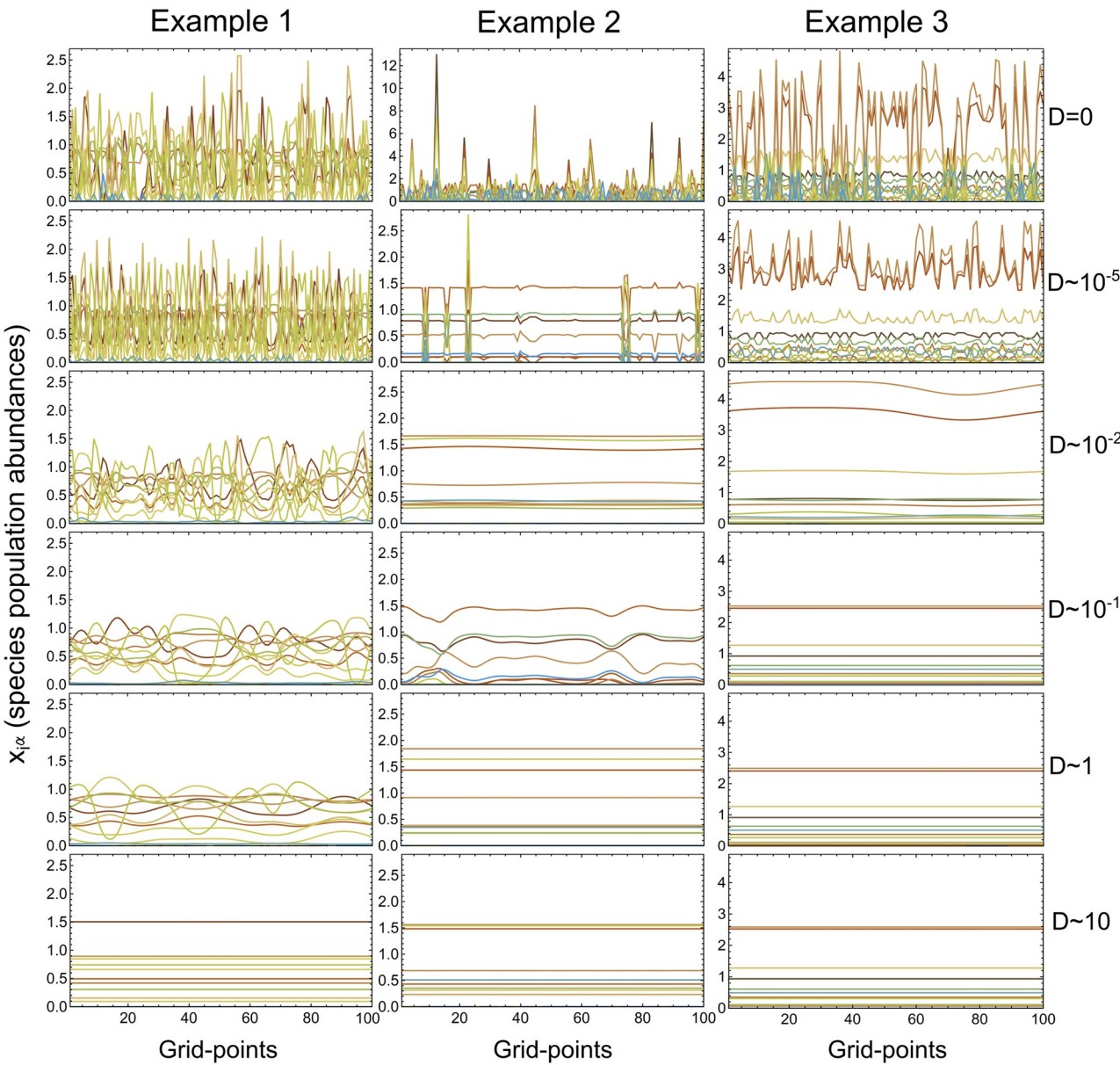

**Fig 2. Spatial structure for varying diffusion rates.** The figure shows three columns of local species abundances at a specific time for each grid-point from three example runs. Diffusion rates (uniformly distributed with $\mu_D \propto \sigma_D$) increase from top to bottom. The trend toward synchronisation of the grid with increasing diffusion magnitudes is clearly seen, although the three systems are seen to synchronise at different magnitudes.

grid-points. The amount of spatial heterogeneity in dynamics and abundances depends on the magnitude of the diffusion constants $D_i$ as shown in Fig 2. If diffusion constants are zero this represents a number of identical disconnected systems with GLV dynamics, while very large constants lead to complete synchronisation of the grid, smoothing out all local differences in the abundances of species. All these local variations at lower diffusion magnitudes lead to a multitude of system characteristics in terms of combinations of grid-point dynamics and phase-shifts.

## Abundance oscillations and crossing stability limits

Species abundance oscillations (global or local) can appear in the entire non-feasible region. We find that in many systems for the spatial GLV, increasing $\sigma$ can make a fixed point transition to a oscillatory pattern with the same diversity. This behaviour is not as prevalent in the non-spatial GLV, which instead commonly switches to another fixed point with a lower species richness, in effect species go extinct if the system is perturbed or pressured. The connected space enables the system to keep its species richness unchanged if perturbed with either global species abundances (species abundances for the meta-populations) oscillating if the grid-points are synchronised or "stay" in a global fixed point by local unsynchronised abundance oscillations. The spatial systems are therefore more robust in terms of extinctions and in the latter case also in global abundances.

If diffusion constants are small so that synchronisation does not take place, and systems are at interaction strengths nearing previous collapse boundaries this means averaging over local species abundances results in global average abundances in accordance with fixed points that are unstable in the non-spatial GLV. Thus, this robustness of the spatial GLV makes it possible for ecosystems with internal heterogeneity to stay stable in parameter regions beyond stability limits of the non-spatial GLV.

In Fig 3 is an example of a system of $N = 20$ maximum biodiversity for a range of $\sigma$ larger than the feasibility limit, and diffusion magnitudes leading to unsynchronised oscillations ($\mu_D = 10^{-2}$, $\sigma_D = 0.4\mu_D$). We see that the unsynchronised local species abundance oscillations lead to stable global species abundances at an interaction strength where there is no fixed point in the non-spatial GLV. Thus the previous stability boundary is crossed by the spatial GLV without chaotic dynamics and almost without any change of the global species abundances.

## Lower variability in abundances

There are different ways the inclusion of a connected space acts to stabilise global and local species abundance oscillations making them less variable (lower amplitudes and/or lower frequencies). The oscillations in the spatial GLV usually reflect a pattern found in the non-spatial GLV, but with the addition of diffusion local abundance oscillations become less violent with lower amplitudes.

Other systems with oscillations might have different "preferred" oscillations for the non-spatial and spatial GLV respectively. In these cases the non-spatial oscillation patterns are higher in amplitudes and sometimes with higher frequencies than in the spatial GLV, see for example the non-spatial GLV oscillations in Fig 3 for one version of such an oscillation pattern. Yet another possibility is that diffusion makes possible a different lower amplitude oscillation pattern not present in the non-spatial GLV, an example of this is shown in Fig 4.

Many of these violent oscillations in the non-spatial system lead to periodic species extinctions, an unrealistic scenario. The spatial GLV on the other hand can avoid global extinctions both by less violent local oscillations leading to less local extinctions as well as recolonisation-extinction dynamics between local areas. Significant lowering of the variability in local species abundances we find when systems are unsynchronised at low to medium diffusion rates. Although such systems commonly have a large spread in local oscillation amplitudes, all grid-points have dampened oscillation amplitudes compared to the non-spatial system and the global abundances are almost constant. When the diffusion rates in systems like these are increased, engendering synchronised oscillations, they usually reflect oscillation patterns present in the non-spatial GLV but again with reduced amplitudes.

In the limit of large diffusion ($\mu_D > 1$ and $\sigma_D = b\mu_D$, $0.3 < b < 0.4$) the ecosystems again retrieve the high amplitudes of the non-spatial GLV oscillations synchronised in all grid-points.

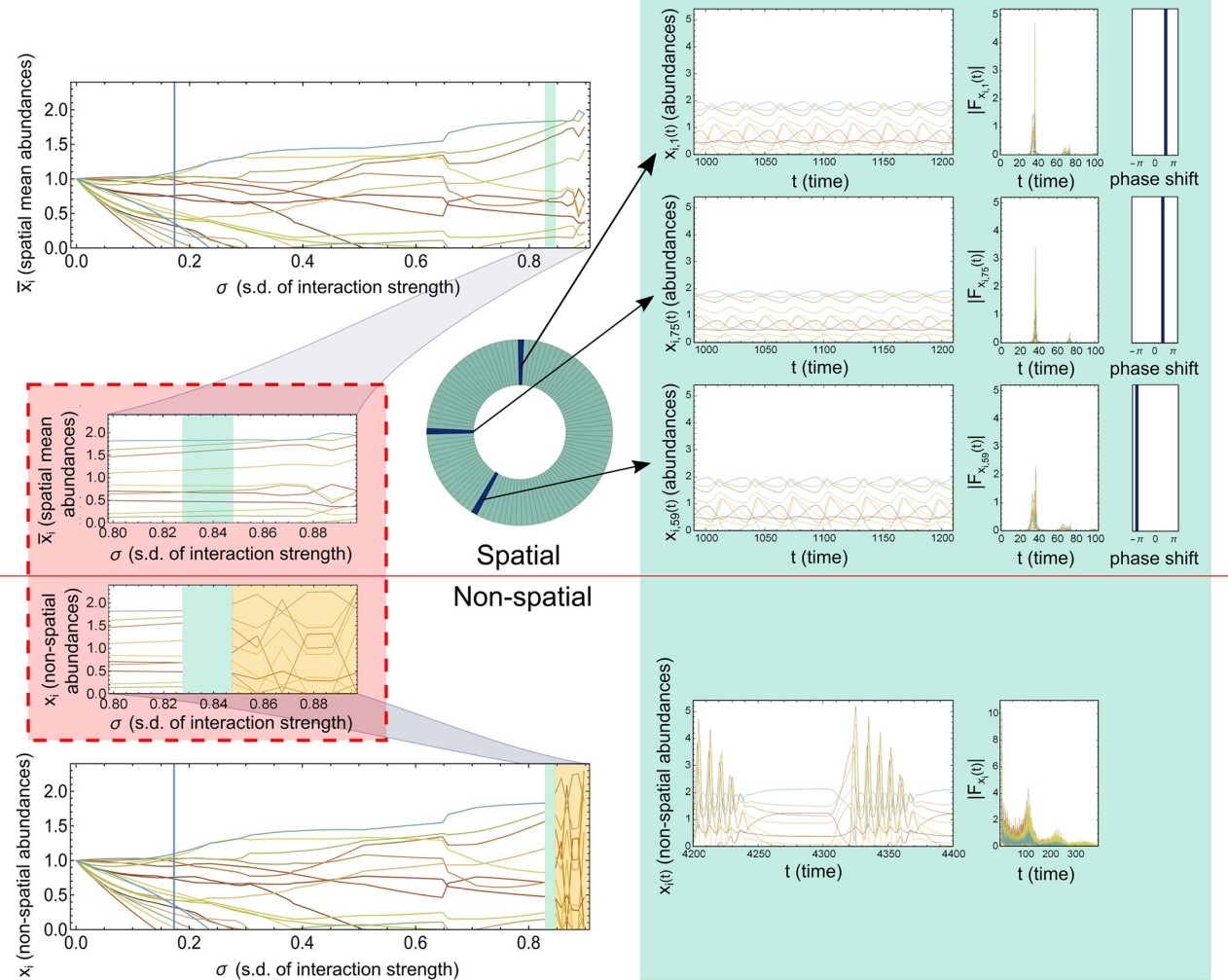

**Fig 3. Unsynchronised local oscillations stabilise.** The figure shows an example of a system of $N = 20$ maximum number of species with out-of-phase oscillations when $\sigma$ is larger than the feasibility limit and diffusion rates are small ($\mu_D = 10^{-2}$, $\sigma_D = 0.4\mu_D$). On the left side on top we show global average fixed point or oscillation species abundances for a system with small diffusion. On the bottom we show fixed point abundances for a system with the same interaction matrix but with no spatial dimension (non-spatial GLV). Both the global average abundances for the spatial system and the non-spatial are shown for $\sigma$ ranging from zero to collapse values with an enlargement of the latter part in the red box. The Green shading in the plots indicate a region where the only stable fixed point for the non-spatial GLV is with 3 species going extinct. On the other hand the global average abundances show no change at all (green area in top left plots). With higher $\sigma$ in the orange region of the non-spatial GLV (bottom left plot) the system is seen to be structurally unstable, while the spatial system on top shows little if any structural instability. These systems behave almost the same with and without a connected space until $\sigma$ is large enough approaching collapse values. To the right on the green background are shown example dynamics for $\sigma$ in the green marked area in the left plots, in different grid-points for the spatial system (top) and for a oscillatory solution for the non-spatial system (bottom). We see in the spatial system oscillatory dynamics in each grid-point example with the same Fourier spectra, but differing phases (the panels to the right). Together the different phases and amplitudes but same frequencies of the local abundance oscillations average to the values corresponding to an unstable fixed point of the non-spatial GLV. For the non-spatial system there is a oscillatory pattern, note however the increase in sharpness in both frequency and amplitude as well as some species going extinct and reappearing, which is not a biologically realistic or stable solution for a ecological system.

Thus when diffusion is increased first oscillations start to synchronise leading to global abundance oscillations and with continued increase in diffusion the species oscillations increase in amplitude until they mimic that of the non-spatial GLV. It is therefore in the middle range of diffusion rates we find the most stabilising effects as seen in Fig 4.

## Species Abundance Variability

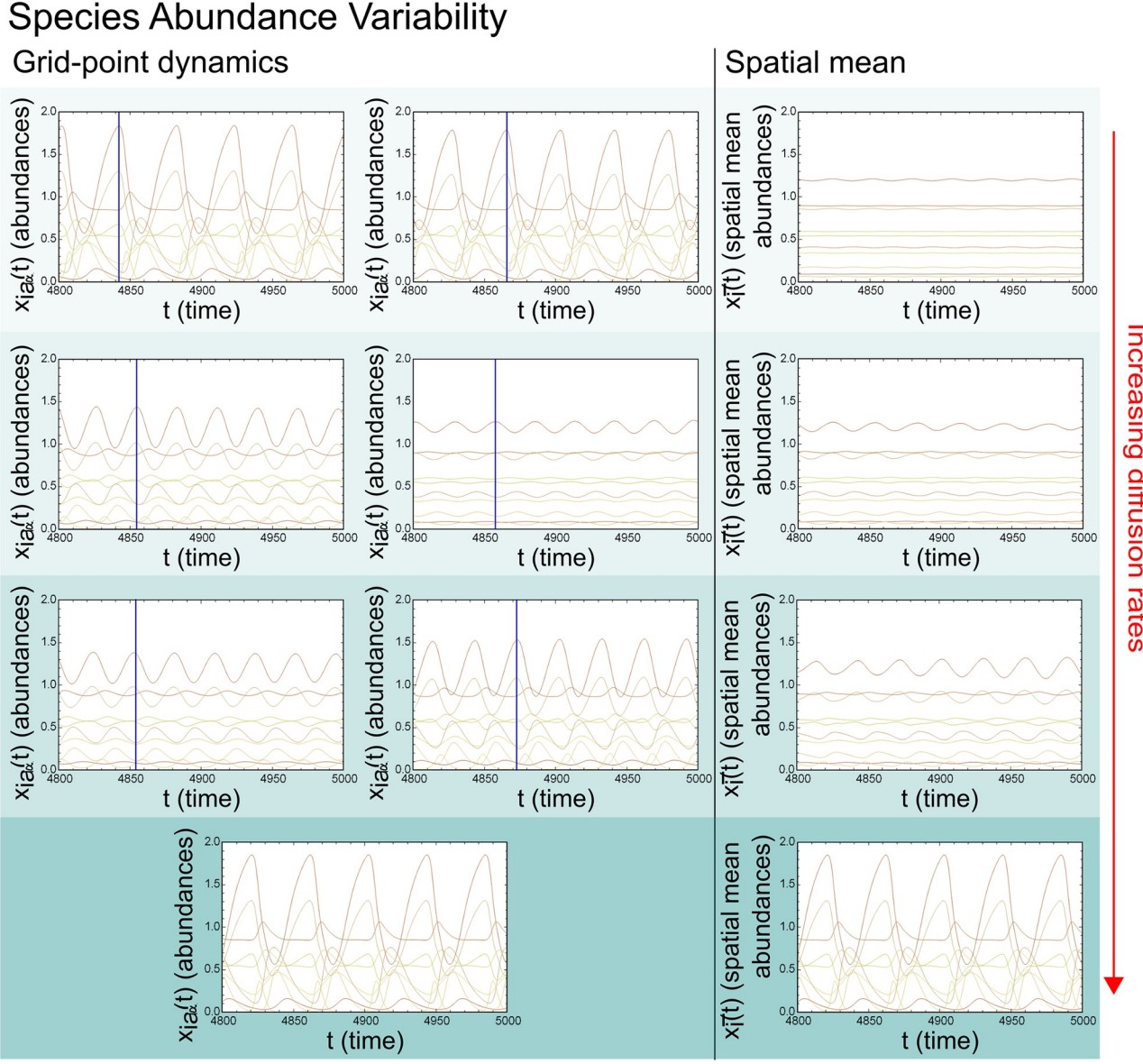

**Fig 4. Local and global species abundances in spatial extended system.** The figure shows an example of a system with varying rates of diffusion from top to bottom $\mu_D = 2.5\sigma_D = 10^{-5}, 10^{-3}, 10^{-1}, 1$ (the green shading increases with increasing diffusion rates). The left column shows the system's dynamics at arbitrary grid-points from high amplitude with low diffusion as the amplitudes are dampened with increasing diffusion and then back to high amplitudes as the diffusion rate is high enough for synchronisation over the grid. If there are two panels in the left column this shows two unsynchronised grid-point dynamics. If there is only one panel the system is synchronised. The right column shows the spatial mean for all grid-points (global average abundances). Note that the unsynchronised oscillations lead to almost constant global species abundances while the synchronised system on the bottom shows violent global oscillations.

### Parameter-space

We have shown examples of dynamics of the spatial GLV and also pointed out the multitude of ecosystem dynamics in a connected space. We can bring some order to this multitude and substantiate our claim that moderate diffusion rates lead to increased stability by gathering statistics on measures that contribute to stability, while varying $\sigma$ and the diffusion rate. The measures we chose for each run are A) Number of grid-points with oscillatory patterns (potentially

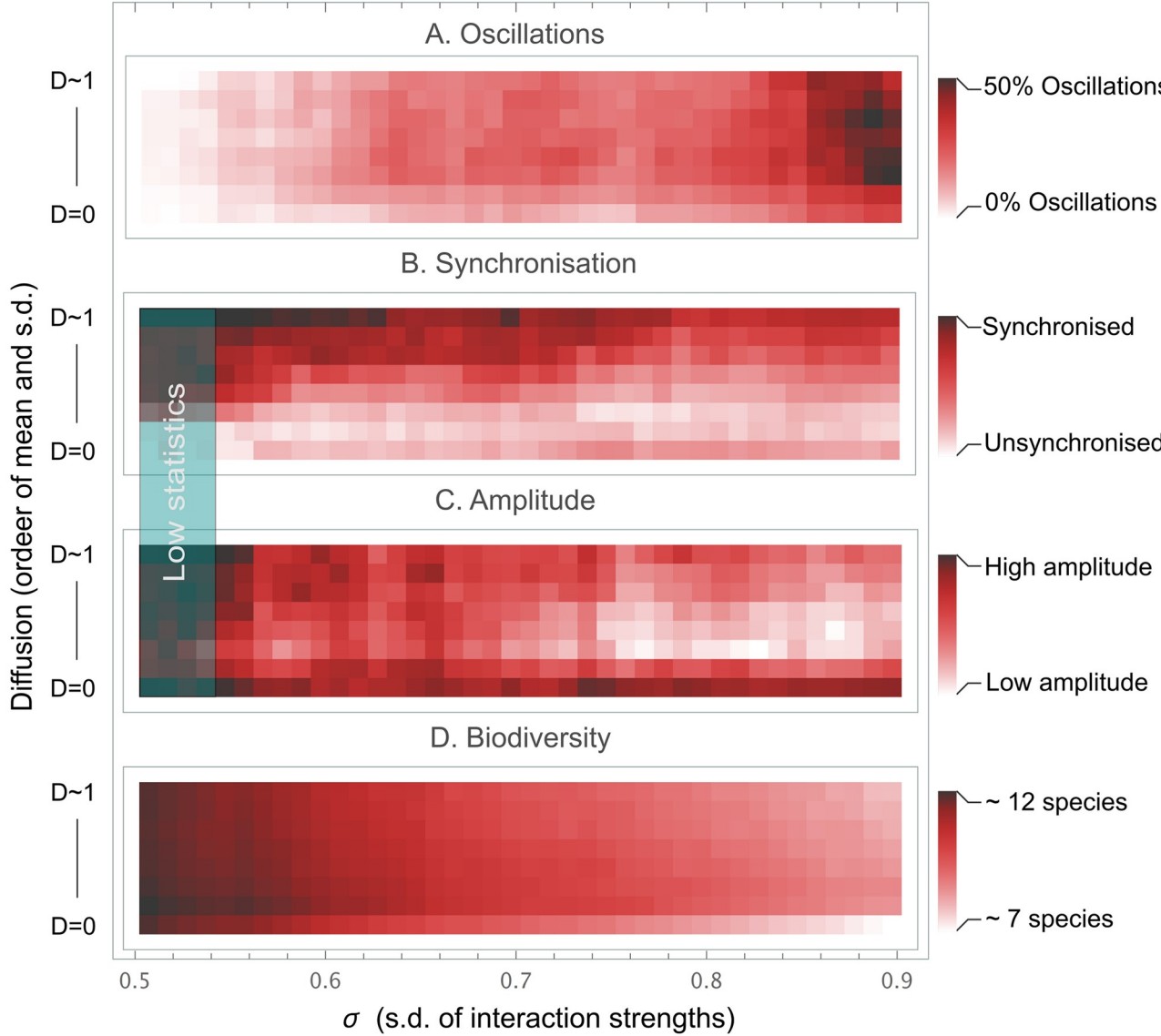

**Fig 5. Parameter-space—Diffusion magnitude vs. s.d. of interaction strength σ.** The figure shows statistics for 70 systems in diffusion magnitude vs. standard deviation parameter-space for panel A) Number of grid-points with oscillatory patterns, panel B) Average maximum phase shift for a species between grid-points if oscillations (degree of synchronisation), panel C) Average amplitude if oscillations, and panel D) Average diversity ($n$). The lowest diffusion in the diagrams is zero, corresponding to completely disconnected space (non-spatial GLV) and the largest $D_i \sim 1$ ($\mu_D = 1$, $\sigma_D = b\mu_D$, $0.3 < b < 0.4$). Worthy of noting is the lower degree of oscillations and the lower diversity in the non-spatial systems. It is also clear that oscillations are present in almost the entire non-feasible region.

different patterns), B) Average maximum phase shift for a species between grid-points if oscillations (degree of synchronisation), C) Average amplitude if oscillations, and D) Average diversity ($n$). The measures are both in combination and one-by-one the basis for different aspects of stability. Statistics over these measures in the non-feasible region for varying magnitude of diffusion rates are shown in Fig 5.

An example of the measures contributing to stability is, the oscillations appearing in almost the entire non-feasible region, although most prominently for higher interaction strengths (panel A), while the average diversity for systems with diffusion is higher than without

diffusion (panel D). This shows the increase in robustness across the entire span of interaction strengths, $\sigma$. Systems can react with local oscillations instead of extinctions if pressured or perturbed. Another example, the lesser fraction of stable fixed points for large $\sigma$ together with less oscillations at $D_i = 0$ for all $i$ (bottom right panel A), show where systems without a connected space start collapsing. Adding moderate rates of diffusion so that the system has a connected space gives rise to unsynchronised oscillations and the ability for systems to stay stable across previous stability limits. Panel C shows the decrease in local species abundance variability (oscillation amplitudes) for the middle range of diffusion magnitudes, where both disconnected and high diffusion lead to violent oscillations. This feature is connected to the degree of synchronisation shown in panel B, where we can see the increase in synchronisation as the diffusion constants are increased in magnitude.

In Fig 6 we summarise the results from the parameter-space in terms of behaviour and stability. The table shows that the middle range of diffusion, in effect systems which allow for a certain amount of dispersal of species between local areas, are the more robust systems over a larger range of interaction strengths.

The diffusion magnitude, as we have shown, we found to have a large impact on dynamics, the variation of diffusion rates between species however had little impact. Different random draws of species diffusion constants with the same magnitude could phase-shift oscillations between runs and introduced a very slight nudge towards asynchronisation, but otherwise introduced no qualitative change in oscillatory patterns or fixed point solutions in the grid-points. The slight nudging towards unsynchronised dynamics could point towards a stabilising effect from large variation in dispersal rates between species, but this was not further investigated.

## Discussion

Ever since May's mathematical argument for the instability of large complex systems [2] challenging the previous view of 'complexity begets stability' advocated by MacArthur [30], it has been an open question if the biodiversity, multitude of interactions between species and the amount of interactions (in effect complexity) of an ecosystem makes it more or less stable. Since then a limit to the complexity an ecosystem can sustain, derived in May's paper and later extended, has persisted. Ecosystems at the edge of this limit or externally pushed so that their complexity exceeds it are predicted to radically change, lose species or collapse. Crucial aspects of such models are that the ecosystems are modelled as isolated homogeneous systems. A resolution to the long standing question of the limit of complexity might not be primarily the structure of interactions but that real ecosystems are both internally heterogeneous and externally connected. Connected ecosystems might gain their capacity of sustaining previously thought to be unstable patterns of cohabitation using the flexibility of local variations and buffer of surrounding ecosystems.

Our model represents an ecosystem or high diversity meta-community for which the abiotic conditions are constant, reflected by our choice of identical interaction strengths, carrying capacities and intrinsic growth rates throughout the connected space. Large ecosystems might include many different types of habitats connected by dispersal, which is mostly found to be positive for biodiversity and robustness [25, 26, 31]. But even in ecosystems with only one type of habitat, the area could be large enough such that external perturbations are not equal, there might be boundaries artificial or natural, or aggregation of species due to conspecific attraction [32, 33], leading to patches with equal species interactions connected by dispersal. As we have shown, in such ecosystems the mere possibility of differential species abundances at different locations enhances stability and diversity.

|  | Low diffusion | Medium diffusion | High diffusion |
|---|---|---|---|
| **Low interaction strength** | Stable fixed points | Stable fixed points | Stable fixed points |
| **Medium interaction strength** | Stable fixed points<br><br>Oscillations:<br>unsynchronised<br>volatile | Stable fixed points<br><br>Oscillations:<br>unsynchronised<br>low amplitudes | Stable fixed points<br><br>Oscillations:<br>synchronised<br>volatile |
| **High interaction strength** | Collapse | Oscillations:<br>unsynchronised<br>low amplitudes | Oscillations:<br>synchronised<br>volatile |

**Fig 6. Stability of spatial GLV.** The table shows in text the type of dynamics present for low, medium and high diffusion rates and s.d of interaction strength. Green indicates stable robust systems and the darker the colour the more robust. Red on the other hand are collapsed or unstable volatile systems. Note that the middle range is green for all s.d of interaction strengths.

The classical stability boundaries have long been known to be a transition from a stable equilibrium fixed point to some qualitatively different dynamics in terms of abundance oscillations, chaos or drop to substantially smaller community [2, 9, 34]. Thus oscillations have been known to be a possible dynamics resulting from crossing the stability boundary. In a connected space the amplitudes of such oscillations are dampened. In addition, if the dispersal rates between patches are low enough, oscillations in local species abundances can be out of phase leading to global abundances remaining largely unchanged. This result is a spatial extension of the time-average effect, which is that time-averages of fluctuating species abundances

in an ecosystem are nearly constant [35, 36]. We can then speculate that the "Portfolio effect", keeping ecosystem properties by unsynchronised oscillations of functionally equal species, is an additional aspect of robustness gained with spatial extension.

The effect of local stabilising oscillations is as we have shown not only present when nearing collapse, but can also prevent extinctions when the system is perturbed. The non-spatial GLV is structurally unstable in the non-feasible region, meaning small structural changes (small increase in $\sigma$) can lead to species extinctions. The connected space, on the other hand can stabilise by local oscillations, thus avoiding the irreversible event of an extinction. With real ecosystems more alike the spatial GLV, we can expect ecosystem to be more robust than many models might suggest [37, 38].

One of the reported mechanisms for high diversity in spatial GLV models with spatial heterogeneity in interactions is that some local habitats function as sources thereby preventing some species from extinction [26]. In our study, this is not a possible mechanism for high diversity, instead we find that spatial connectivity both promotes oscillatory patterns with higher diversity than fixed points as well as allowing for a larger spectrum of the possible dynamics some of which suffer less extinction and thus contribute to higher global diversity. A similar mechanism is found in chaotic dynamics, where it is reported that booms and bust (asynchronous abundances) at different identical locations nurture persistent high diversity dynamics [27].

Adding a connected space increases the dynamics available for a system. This is apparent in our study as well as in [28], studying the abruptness between such different dynamical realisations along a spatial gradient. We find that not only is there room for different combinations of dynamics in the grid but the connected space can also facilitate dynamics not found or unstable in spatially unresolved systems. This is a source of stability, but we also argue that the increased dynamical possibilities is itself a mechanism of robustness for the system. Importantly we also find that this repertoire of dynamics is largest at intermediate levels of dispersal, an observation in line with previous studies [21, 25, 26, 39].

It is quite remarkable that the inclusion of space even with homogeneous patches and without external forcing is sufficient for dynamic spatial heterogeneity to emerge in the GLV, including regimes of stable global abundances and persistent local abundance fluctuations. High diversity meta-communities have been shown to also display local species turn-over in absence of external influence [40]. The fact that undisturbed high diversity meta-communities have such a wealth of dynamical behaviour can have a large impact on biological monitoring. To correctly asses and manage ecosystems it is vital that we can predict what behaviour can be expected from intrinsic system dynamics so that we do not wrongfully assign such dynamics to environmental change or anthropogenic pressures and invest our efforts where it is uncalled for.

With the help of spatially extended models many novel mechanisms promoting diversity and stability have been found and intrinsic dynamics uncovered, but there are avenues relevant to real ecosystems still unexplored. In this study we used different diffusion constants for different species but all within the same order of magnitude, with no definite qualitative change in stability properties compared to constant diffusion rates. It is still unclear how a larger spread in species mobility, larger spread in diffusion constants, would effect stability. This, both because of the very slight nudge towards unsynchronised dynamics and since spatial patterning in reaction diffusion systems tend to appear in systems where the components have a large spread in diffusion constants. We can speculate that spatial species patterning might appear in such systems. Varying the diffusion constants between grid-points is also a direction left unexplored. Varying constants can represent varying distance, some natural obstacle, a road or systems with internal dynamics connected to surrounding ecosystems. With varying

diffusion constants we can therefore both study "unsymmetrical" habitats where interactions between local areas differ as well as how local disturbances might affect an ecosystem and connected ecosystems.

Left for future work is also varying the standard deviation of interaction strength $\sigma$ over the grid. The inverse interaction strength can be interpreted as a proxy for available habitat, since a reduction of available space will force the existing species closer together and to interact more which is an increase in $\sigma$. Thus, with different $\sigma$ we can model a grid with different local habitat sizes as well as a system's response to local habitat losses if perturbing one or some of the $\sigma$ at a time.

## Conclusion

A large ecosystem can have internal spatial heterogeneity in local species abundances. Although, many insights into ecosystems' functioning and stability have been gained through studies where such heterogeneity is averaged out, in this study we find that adding this possibility promotes ecosystem robustness and diversity. We add the possibility of abundance heterogeneity by modelling an ecosystem as collection of communities connected by dispersal. In this spatially resolved system a limit of complexity for stability is no longer a limit to qualitative change or collapse. With the connected space acting as a buffer harbouring unsynchronised local abundances, global abundances are kept constant beyond stability limits of spatially homogeneous systems. We also found that spatially extended systems have lower variability in abundance fluctuations and the ability to avoid extinctions by local species oscillations, thus promoting high diversity. The increase in dynamical possibilities and combinations is a source of robustness for the global system. This repertoire of dynamics is maximal at intermediate dispersal which we therefore find to promote the most robust ecosystems.

## Author Contributions

**Conceptualization:** Susanne Pettersson, Martin Nilsson Jacobi.

**Formal analysis:** Susanne Pettersson, Martin Nilsson Jacobi.

**Investigation:** Susanne Pettersson, Martin Nilsson Jacobi.

**Software:** Susanne Pettersson.

**Validation:** Susanne Pettersson.

**Visualization:** Susanne Pettersson.

**Writing – original draft:** Susanne Pettersson.

**Writing – review & editing:** Susanne Pettersson, Martin Nilsson Jacobi.

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
