## [Decision Letter · Decision Letter 0]

1 Jun 2021

Dear Miss Pettersson,

Thank you very much for submitting your manuscript "Spatial heterogeneity enhance robustness of large multi-species ecosystems" for consideration at PLOS Computational Biology.

As with all papers reviewed by the journal, your manuscript was reviewed by members of the editorial board and by several independent reviewers. In light of the reviews (below this email), we would like to invite the resubmission of a significantly-revised version that takes into account the reviewers' comments.

Thank-you for your submission and apologies for the delay in providing a recommendation.

Both reviewers see the question of how spatial mechanisms influence both stability and robustness as an extremely important area of investigation. However, both reviewers also question whether there is a sufficient amount of novel insight (relative to existing literature) in the current manuscript to merit publication. Reviewer 1 highlights specific recent papers combining pairwise species interactions and spatial processes---I agree with the reviewer that it is important to highlight what precisely are the differences and novel findings here so that readers will understand what is being added to our understanding. Reviewer 2 raises more general concerns about the broader context of diversity-stability relationships. Like the reviewer, I am not sure that the goals here are identical to those papers (and perhaps they speak more to the question of robustness in the current ms than to local stability) but it is fair to say that readers may find this context extremely helpful.

In seeing these concerns, I believe that revisions may be difficult. I do think that it would be possible to send the paper out to review again if the authors were to choose to resubmit to PLOS Comp Biol, but the authors would need to highlight more clearly the comparison with existing results and broader concepts.

We cannot make any decision about publication until we have seen the revised manuscript and your response to the reviewers' comments. Your revised manuscript is also likely to be sent to reviewers for further evaluation.

Sincerely,

Natalia L. Komarova

Deputy Editor

PLOS Computational Biology

Natalia Komarova

Deputy Editor

PLOS Computational Biology

Thank-you for your submission and apologies for the delay in providing a recommendation.

Both reviewers see the question of how spatial mechanisms influence both stability and robustness as an extremely important area of investigation. However, both reviewers also question whether there is a sufficient amount of novel insight (relative to existing literature) in the current manuscript to merit publication. Reviewer 1 highlights specific recent papers combining pairwise species interactions and spatial processes---I agree with the reviewer that it is important to highlight what precisely are the differences and novel findings here so that readers will understand what is being added to our understanding. Reviewer 2 raises more general concerns about the broader context of diversity-stability relationships. Like the reviewer, I am not sure that the goals here are identical to those papers (and perhaps they speak more to the question of robustness in the current ms than to local stability) but it is fair to say that readers may find this context extremely helpful.

In seeing these concerns, I am suggesting a rejection. I do think that it would be possible to send the paper out to review again if the authors were to choose to resubmit to PLOS Comp Biol, but the authors would need to highlight more clearly the comparison with existing results and broader concepts.

Reviewer's Responses to Questions

**Comments to the Authors:**

Reviewer #1: The authors look at dynamics of interacting species (via the GLV equations), and add connected space to the system. The species interact in each location and are able to migrate between the different locations. They find that stability limits on GLV system when considered without space, are no longer restrictive due to the possibility of spatial heterogeneity, and the related appearance of unsynchronized oscillations of the abundances.

The manuscript looks well-written and draws clear and potentially interesting conclusions. But at the moment it is hard for me to judge the novelty of the work because it seems, as far as I can see, very closely related to the following publications:

* Roy, Felix, Matthieu Barbier, Giulio Biroli, and Guy Bunin. “Complex Interactions Can Create Persistent Fluctuations in High-Diversity Ecosystems.” PLOS Computational Biology 16, no. 5 (2020): e1007827.

* Pearce, Michael T., Atish Agarwala, and Daniel S. Fisher. “Stabilization of Extensive Fine-Scale Diversity by Ecologically Driven Spatiotemporal Chaos.” Proceedings of the National Academy of Sciences 117, no. 25 (2020): 14572–83.

In both these references, different locations in space with GLV (or replicator equations) are coupled via migration. Dynamics of the abundances appear and are argued to allow for stability bounds to be crossed. The theme of de-synchronization plays a role in these references as well, as does the non-trivial dependence on migration rates.

One difference between the present manuscript and these references, is that the locations in space are here placed on a grid versus all-to-all connections. But in the present manuscript the topology of the connections is claimed not to be very important to the qualitative results (when using 1d versus 2d or 3d grid).

It may very well be that the present manuscript contains novel results beyond these publications, but in order to make a judgement, I'd first like to see a comparison by the authors, between their work and the above references.

Reviewer #2: Please find my comments attached

**Have all data underlying the figures and results presented in the manuscript been provided?**

Reviewer #1: Yes

PLOS authors have the option to publish the peer review history of their article (what does this mean?). If published, this will include your full peer review and any attached files.

Reviewer #1: No

Reviewer #2: No

**Have the authors made all data and (if applicable) computational code underlying the findings in their manuscript fully available?**

Reviewer #2: **No: **I see no working links to data or code repositories
---

## [Decision Letter · Decision Letter 1]

25 Aug 2021

Dear Miss Pettersson,

Thank you very much for submitting your manuscript "Spatial heterogeneity enhance robustness of large multi-species ecosystems" for consideration at PLOS Computational Biology. As with all papers reviewed by the journal, your manuscript was reviewed by members of the editorial board and by several independent reviewers. The reviewers appreciated the attention to an important topic. Based on the reviews, we are likely to accept this manuscript for publication, providing that you modify the manuscript according to the review recommendations.

Thanks for your work on addressing the reviewer comments. I would be happy to consider a revised manuscript that addresses the remaining (relatively minor) comments of reviewer 2

Sincerely,

James O'Dwyer

Deputy Editor

PLOS Computational Biology

Natalia Komarova

Deputy Editor

PLOS Computational Biology

[LINK]

Thanks for your work on addressing the reviewer comments. I would be happy to consider a revised manuscript that addresses the remaining (relatively minor) comments of reviewer 2

Reviewer's Responses to Questions

**Comments to the Authors:**

Reviewer #1: The authors have taken the comments of both myself and the other reviewer into account. The new version of the manuscript better connects with the existing literature on spatial effects, and in particular metacommunities.

The manuscript contains a number of interesting observations, specifically on the conditions under which space can stabilize communities at higher species richness, and the dynamics which allow this to happen.

Reviewer #2: Review uploaded as attachment

**Have the authors made all data and (if applicable) computational code underlying the findings in their manuscript fully available?**

Reviewer #1: None

Reviewer #2: Yes

PLOS authors have the option to publish the peer review history of their article (what does this mean?). If published, this will include your full peer review and any attached files.

Reviewer #1: No

Reviewer #2: **Yes: **Jacob O'Sullivan

Figure Files:

Data Requirements:

Reproducibility:

References:

---

## [Editor Report · Decision Letter 2]

7 Oct 2021

Dear Miss Pettersson,

We are pleased to inform you that your manuscript 'Spatial heterogeneity enhance robustness of large multi-species ecosystems' has been provisionally accepted for publication in PLOS Computational Biology.

Best regards,

James O'Dwyer

Deputy Editor

PLOS Computational Biology

Natalia Komarova

Deputy Editor

PLOS Computational Biology

---

## [Editor Report · Acceptance letter]

21 Oct 2021

PCOMPBIOL-D-21-00445R2 

Spatial heterogeneity enhance robustness of large multi-species ecosystems

Dear Dr Pettersson,

I am pleased to inform you that your manuscript has been formally accepted for publication in PLOS Computational Biology. Your manuscript is now with our production department and you will be notified of the publication date in due course.

With kind regards,

Olena Szabo
